# A general strategy for recycling polyester wastes into carboxylic acids and hydrocarbons

Wei Zeng[1,2], Yanfei Zhao[1,2], Fengtao Zhang[1], Rongxiang Li[1,2], Minhao Tang[1,2], Xiaoqian Chang[1,2], Ying Wang[1,2], Fengtian Wu ©[1], Buxing Han ©[1,2] & Zhimin Liu ©[1,2] ✉

Chemical recycling of plastic wastes is of great significance for sustainable development, which also represents a largely untapped opportunity for the synthesis of value-added chemicals. Herein, we report a novel and general strategy to degrade polyesters via directly breaking the $C_{alkoxy}$-O bond by nucleophilic substitution of halide anion of ionic liquids under mild conditions. Combined with hydrogenation over Pd/C, 1-butyl-2,3-dimethylimidazolium bromide can realize the deconstruction of various polyesters including aromatic and aliphatic ones, copolyesters and polyester mixtures into corresponding carboxylic acids and alkanes; meanwhile, tetrabutylphosphonium bromide can also achieve direct decomposition of the polyesters with $\beta$-H into carboxylic acids and alkenes under hydrogen- and metal-free conditions. It is found that the hydrogen-bonding interaction between ionic liquid and ester group in polyester enhances the nucleophilicity of halide anion and activates the $C_{alkoxy}$-O bond. The findings demonstrate how polyester wastes can be a viable feedstock for the production of carboxylic acids and hydrocarbons.

Plastics, once termed as the materials of 1000 uses, are ubiquitous and continue to expand progressively in human production and life since discovered for the first time in 1907[1]. Plastics have brought great convenience, but massive plastic wastes have caused severe harm to the ecological environment and human beings owing to the difficulty in degradation[2–4]. Though some biodegradable plastics such as poly-lactic acid (PLA) and poly(butylene adipate-co-terephthalate) (PBAT) have been developing rapidly in recent years, they are eventually degraded into $CO_2$, $CH_4$, and $H_2O$ in the natural environment, which intensifies the greenhouse effect and does not form the carbon closed-loop indeed[5]. Therefore, a circular economy in which spent plastics are recycled and repurposed is urgently needed[6]. Four major approaches have been developed for plastics recycling: closed-loop recycling, mechanical recycling, chemical recycling, and energy recovery (incineration)[7,8]. Among these approaches, chemical recycling and upcycling not only provide efficient ways to eliminate the environmental pollution of plastic wastes but also offer new routes to access valuable chemicals, which thus has attracted much attention[9–13]. To date, chemical recycling strategies for polyester plastics including pyrolysis[14], hydrolysis[15], alcoholysis[16], ammonolysis[17], hydrogenolysis[18], and hydrosilylation[19] have been reported, which can produce corresponding carboxylic acids, esters, amides, and alcohols, etc. Especially, the degradation of poly(ethylene terephthalate) (PET) and PLA has been widely investigated, while that of other polyesters has been seldom reported[20].

Carboxylic acids are a kind of important chemicals extensively applied in the industry, with a global market share of $10.12 billion in 2021, which is predicted to be around $13.04 billion by 2026[21,22]. Hydrocarbons are important fuels and chemical raw materials, which are mainly derived from petroleum refining. The hydrolysis of

[1]Beijing National Laboratory for Molecular Sciences, CAS Laboratory of Colloid and Interface and Thermodynamics, CAS Research/Education Center for Excellence in Molecular Sciences, Center for Carbon Neutral Chemistry, Institute of Chemistry, Chinese Academy of Sciences, 100190 Beijing, China. [2]University of Chinese Academy of Sciences, 100049 Beijing, China. ✉e-mail: liuzm@iccas.ac.cn

polyesters can produce carboxylic acids and alcohols via the cleavage of the ester $C_{acyl}$-O bond. However, it generally suffers from limited selectivity and serious difficulty in separation due to the generation of oligomeric byproducts and the use of excessive solvents. From the chemical structures of the polyesters, it is obvious that the breakage of the ester $C_{alkoxy}$-O bond in polyesters can remain in the -COO- group, which may thus result in the formation of carboxylic acids exclusively. This has been supported by the hydrogenolysis of poly(ethylene terephthalate) (PET) into terephthalate (TPA) and ethane at high temperatures[23,24].

Ionic liquids (ILs) composed of organic cations and organic/inorganic anions have been emerging as a kind of functional materials and applied in various chemical processes due to their unique properties[25–28]. It was reported the basic ILs (e.g., [EMIm][AcO]) could accomplish PHB decomposition into crotonic acid[29,30]. Furthermore, the ILs with halide anions have been widely employed as solvents or catalysts in various chemical reactions. For example, 1-butyl-3-methylimidazolium chloride ([BMIm]Cl) as solvent could efficiently dissolve cellulose because of the strong hydrogen-bonding interaction[31], and combined with metal halides (e.g., $CrCl_2$) it could convert sugars (e.g., fructose, glucose) into 5-hydroxymethylfurfural in high yield[32]. Lewis acidic N-butylpyridinium chloride-aluminum chloride ([$C_4Py$]Cl-2$AlCl_3$) could achieve upcycling of polyolefins into liquid alkanes through tandem cracking-alkylation process at temperatures below 100 °C[33].

In this work, we discovered that the ILs with halide anions could break the ester $C_{alkoxy}$-O bond in polyesters via the nucleophilic substitution by the halide anions of the ILs, and thus proposed a general strategy to degrade polyesters into carboxylic acids and hydrocarbons, as shown in Fig. 1. Combined with hydrogenation over Pd/C, the IL could realize the decomposition of various polyesters including aromatic and aliphatic ones, copolyesters and polyester mixtures into corresponding carboxylic acids and alkanes. In the meantime, it could also achieve direct deconstruction of the polyesters with $\beta$-H bond into carboxylic acids and alkenes under metal-free conditions. This universal strategy achieves the complete degradation of polyesters into carboxylic acids with atomic economy. Furthermore, the production of hydrocarbons simplifies the separation of products.

## Results

### Screening ionic liquids

Taking polyglycolic acid (PGA) as a representative of polyesters, in the presence of Pd/C various ILs were screened under the $H_2$ atmosphere of 5 MPa and 180 °C (Supplementary Fig. 3). It was found that only the ILs with halide anions that possess nucleophilicity could achieve the PGA degradation, producing acetic acid. This indicates that the catalytic system achieves the breakage of the ester $C_{alkoxy}$-O bond in PGA, which is caused by the nucleophilic attack of the halide anion on the $C_{alkoxy}$ atom. Among the tested ILs, 1-butyl-2,3-dimethylimidazolium bromide ([BMMIm]Br) showed the best performance, affording an acetic acid yield of 95% (Fig. 2). By comparison, it could be deduced that the activity of these ILs is related to the nucleophilicity of their anions, which is greatly influenced by the synergy of the electrostatic interaction and hydrogen-bonding interaction[34] between the IL cation and anion. As known, the nucleophilicity index of nucleophiles could reflect the degree of nucleophilicity[35]. We estimated the nucleophilicity index of some ILs via density function theory (DFT) calculations[36] (Supplementary Fig. 4), and found that the IL that displayed high activity has a high nucleophilicity index. The effects of temperature, pressure, time, and the amounts of IL on the PGA decomposition were also investigated (Supplementary Fig. 5), and it was demonstrated that temperature influenced the PGA degradation more obviously. It should be pointed out that the PGA degradation could proceed completely, but it was difficult to collect all the generated acetic acid owing to its high volatility.

### Recycling polyesters over [BMMIm]Br-Pd/C under the $H_2$ atmosphere

We selected [BMMIm]Br-Pd/C as the catalytic system for the degradation of various polyesters under the $H_2$ atmosphere (Fig. 2). It was found that the biodegradable polyesters including PLA and PHB were decomposed into propionic acid and butyric acid, respectively, which are different from their hydrolysis products. Expectedly, poly(D, L-lactide-co-glycolide) (PLGA), as the lactide and glycolide co-polyester, was deconstructed into acetic acid and propionic acid.

Besides, other polyesters including PET, poly(butylene terephthalate) (PBT), poly(1,4-butylene succinate) (PBS), poly(1,4-butylene adipate) (PBA), poly(butylene adipate-co-terephthalate) (PBAT) and polyester mixtures could be decomposed over this catalytic system as well, producing corresponding carboxylic acids and alkanes. PET, as a representative of aromatic polyesters and the most widely investigated polyester, was examined, which was converted into its starting monomer terephthalate acid (TPA) with ethane generation at 180 °C. Furthermore, the feedstock from a PET plastic bottle could also be successfully converted into TPA, and a comparable yield was achieved using a smaller amount of IL in toluene. These findings demonstrate that the recycling of PET over the [BMMIm]Br-Pd/C catalytic system has a promising application prospect.

**Fig. 1 | A general strategy for recycling polyesters into carboxylic acids and hydrocarbons.** IL as catalyst, $R_1$ = H or $CH_3$, $R_2$ = aryl or alkyl groups, X = halogen, $m \geq 0$, $m_1 \geq 1$, $m_2 \geq 2$.

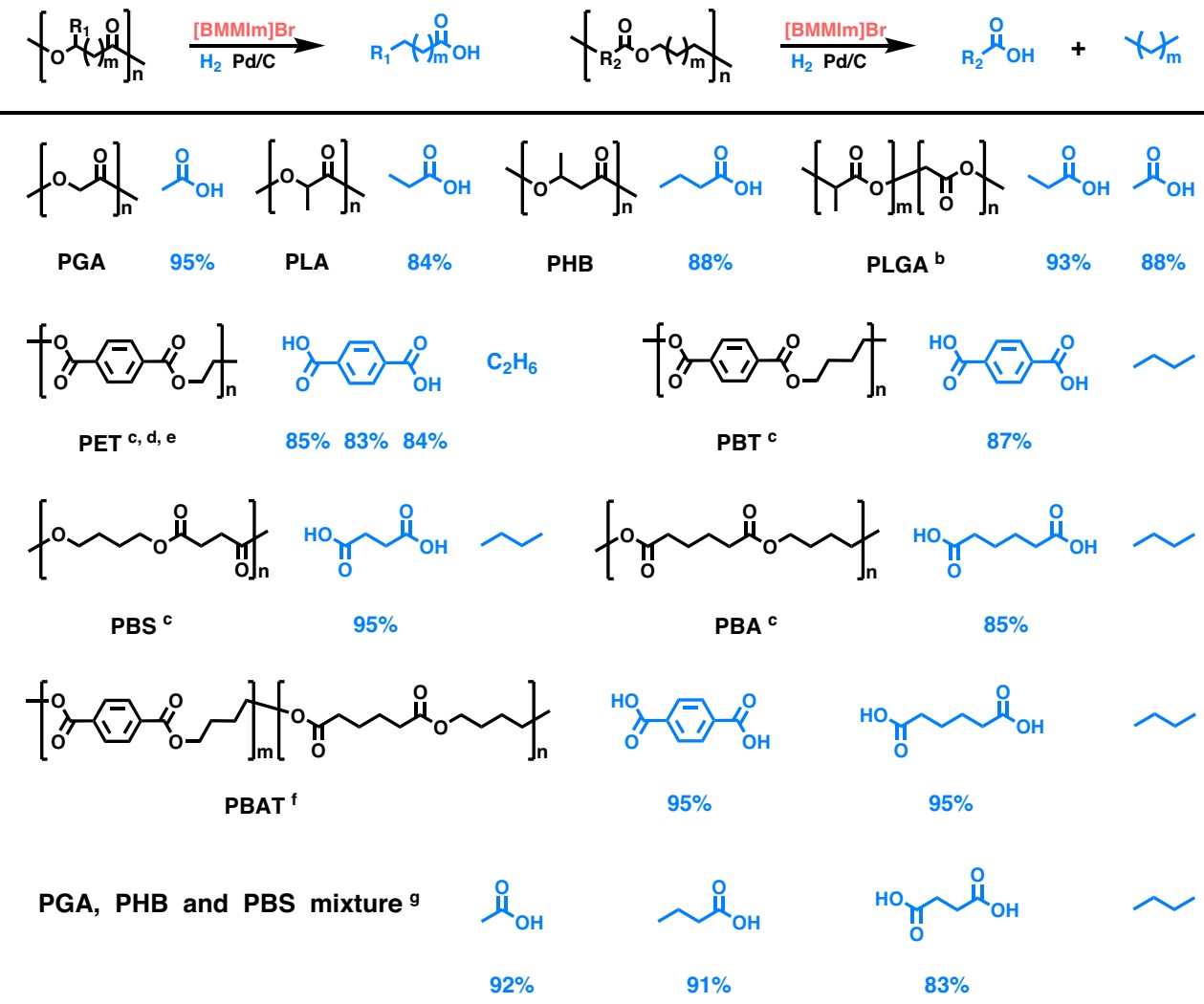

**Fig. 2 | Decomposition of various polyesters (R₁ = H or CH₃, R₂ = aryl or alkyl groups, m ≥ 0) over [BMMIm]Br-Pd/C under the H₂ atmosphere.** ᵃReaction conditions: **a** polyester (1 mmol), [BMMIm]Br (2 mmol), Pd/C (5 mg, 5 wt% Pd), 5 MPa H₂, 180 °C, 24 h; **b** polyester (0.5 mmol); **c** polyester (0.4 mmol), 48 h; **d** PET bottle (0.4 mmol), [BMMIm]Br (3 mmol), 48 h; **e** PET bottle (0.4 mmol), [BMMIm]Br (1 mmol), toluene (1 mL), 48 h; **f** polyester (0.2 mmol), 48 h; **g** PGA (0.5 mmol), PHB (0.5 mmol), PBS (0.2 mmol), 48 h.

Similarly, aromatic polyester PBT was also decomposed into the starting monomer TPA in an excellent yield, accompanied by butane. PBS, which is derived from succinic acid and 1,4-butanediol and as a representative of polyesters from the coupling of aliphatic diacids and diols, was likewise degraded into the starting monomer succinic acid and butane. Similar to PBS, PBA was also converted into adipic acid and butane. Furthermore, PBAT, a representative of aromatic-aliphatic copolyesters, was deconstructed into TPA, adipic acid, and butane.

To explore the applicability of this catalytic system in practice, the degradation of a polyester mixture composed of PGA, PHB, and PBS was performed, and all the polyesters were successfully decomposed, producing acetic acid, butyric acid, and succinic acid, accompanied by butane as expected. This suggests that the [BMMIm]Br-Pd/C catalytic system is applicable for degrading the spent polyester mixtures, which could effectively overcome the difficulty in the separation of the collected polyester plastics. Remarkably, the generation of alkanes in the above processes could contribute to driving the degradation of polyesters to proceed thoroughly and make the separation of carboxylic acids from the reaction systems easier. Besides, in the above processes to degrade polyesters, we found that the melting points and appearances of polyesters obviously influence the polyester decomposition process (Supplementary Fig. 6).

## Mechanistic studies on recycling polyesters over [BMMIm]Br-Pd/C under the H₂ atmosphere

To explore the roles of the IL catalyst in degrading polyester, taking methyl benzoate (MB) as a model of the structural unit of polyesters and 1-butyl-2,3-dimethylimidazolium chloride ([BMMIm]Cl) as a representative of the ILs, high-temperature NMR experiments including ¹H, ¹³C, ¹⁷O and ³⁵Cl NMR tests were performed at 90 °C. Compared to that of pure IL, the ³⁵Cl NMR spectra of the IL in the mixture changed obviously, with the ³⁵Cl resonance signal shifting from 58.64 ppm for pure IL to 54.35 ppm (Fig. 3A). This suggests that there exists a strong interaction between the IL and MB, which weakens the electrostatic force between the IL cation and anion, and thus increases the electron cloud density and the nucleophilicity of the Cl⁻ anion. From the ¹⁷O NMR spectra (Fig. 3B), it is clear that the resonance signal assigned to the carbonyl O atom of MB shifted from 341.75 ppm for pure MB to 342.02 ppm as MB mixed with the IL, which reflects a decrease in electron cloud density of the carbonyl O atom. However, the ¹H and ¹³C NMR spectra of the IL cation remained almost unchanged as the IL mixed with MB (Supplementary Fig. 7), indicating hardly changed electron cloud density of the IL cation. Therefore, it can be deduced that the increased electron cloud density of the IL anion may originate from the shift of the electron cloud from MB to the IL anion

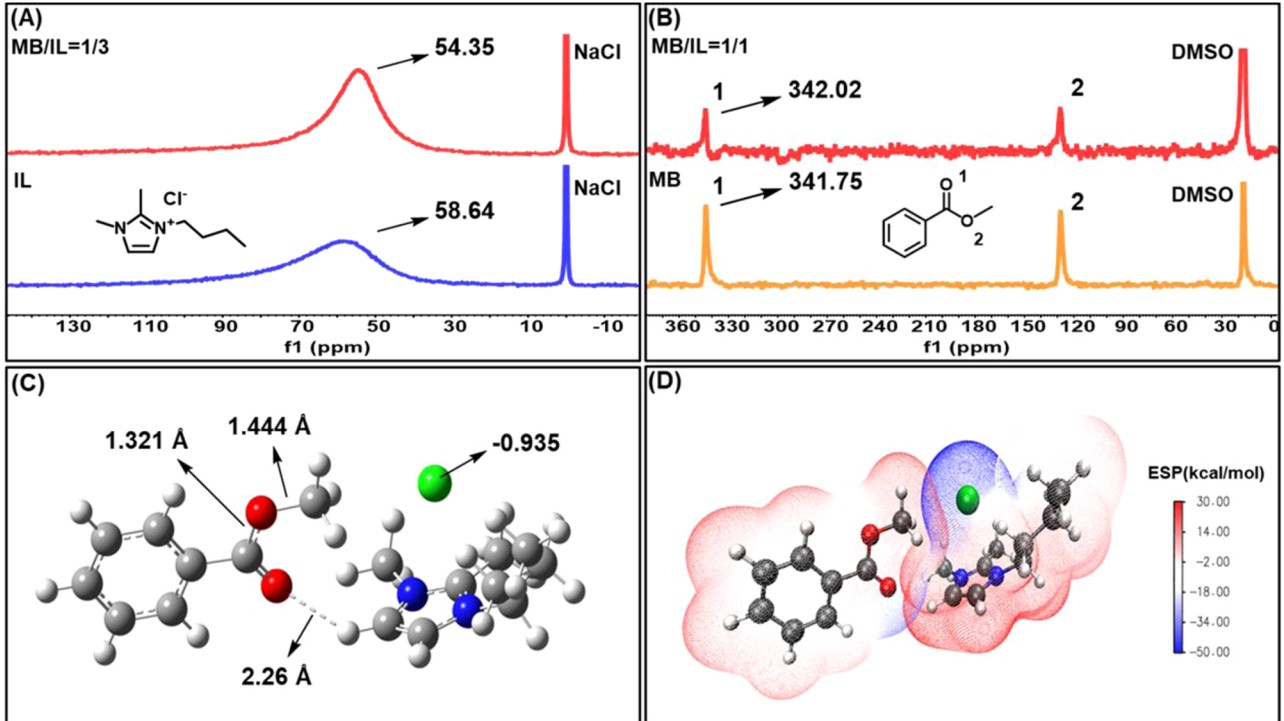

**Fig. 3 | High-temperature NMR experiments and DFT calculations. A** $^{35}$Cl NMR spectra of pure [BMMIm]Cl and the MB-[BMMIm]Cl mixture were collected at 90 °C. **B** $^{17}$O NMR spectra of pure MB and the MB-[BMMIm]Cl mixture were collected at 90 °C. **C** Optimized interaction geometry of the MB-[BMMIm]Cl complex. **D** Electrostatic potential (ESP) distribution of the MB-[BMMIm]Cl complex. (white ball: H; black ball: C; blue ball: N; green ball: Cl).

via the IL cation. Based on the above NMR analysis results, we could reasonably speculate that the carbonyl O atom of MB may form a hydrogen bond with the H atom in the imidazolium ring of the IL cation[37], which effectively weakens the electrostatic interaction between the IL cation and anion, thus enhancing the nucleophilicity of the IL anion.

DFT calculations provide more information on hydrogen bonds between the IL cation and ester group[36]. From the optimized interaction geometry of the MB-[BMMIm]Cl complex, the atomic distance between the carbonyl O atom in methyl benzoate (MB) and the C4-H atom of the imidazolium ring in IL was calculated to be 2.26 Å (Fig. 3C), suggesting the formation of the hydrogen bond between MB and the IL cation[38]. Moreover, the electrostatic potential (ESP) distribution of the MB-IL complex clearly shows that the negative surface potential (blue area) of the carbonyl O atom in MB overlapped with the positive surface potential (red area) of the C4-H atom in IL cation (Fig. 3D), which thus also confirms the formation of the hydrogen bond. Besides, the NBO charges of the Cl$^-$ anion in pure IL and the MB-IL complex were estimated to be −0.916 and −0.935, respectively, meaning that the Cl$^-$ anion becomes more negative in the MB-IL complex (Fig. 3C and Supplementary Fig. 8B). This variation supports the enhanced nucleophilicity of the IL anion, which is in line with the results of NMR analyses. It further confirms that the hydrogen bond formed between MB and IL weakens the interaction between the IL cation and anion.

To explore possible reaction pathway of polyester degradation, taking methyl benzoate (MB) as a model ester for the structural unit of polyesters, control experiments were carried out under catalysis of [BMMIm]Br and Pd/C in the presence of H$_2$ or N$_2$. As expected, MB was converted into benzoic acid and methane under the H$_2$ atmosphere (Fig. 4A and Supplementary Fig. 9), while it was transformed into [BMMIm][PhCOO] and methyl bromide under the N$_2$ atmosphere (Fig. 4B and Supplementary Fig. 10). These results demonstrate that the Br$^-$ anion has nucleophilicity and directly attacks the C$_{alkoxy}$ atom of the ester group rather than the C$_{acyl}$ atom in MB, thus resulting in the

cleavage of the C$_{alkoxy}$-O bond and the formation of bromide. Furthermore, 3-bromobutyric acid could be hydrogenated into butyric acid over the [BMMIm]Br-Pd/C catalytic system under the H$_2$ atmosphere (Fig. 4C and Supplementary Fig. 11).

On the basis of the aforementioned experimental results and discussion, a plausible reaction pathway is proposed for the decomposition of polyesters over the [BMMIm]Br-Pd/C catalytic system under the H$_2$ atmosphere, as illustrated in Fig. 4D. The breakage of the ester C$_{alkoxy}$-O bond is the key, which is achieved via the nucleophilic attack of the Br$^-$ anion on the C atom of the C$_{alkoxy}$-O bond, generating two intermediates: bromide and carboxylate anion that is stabilized by the IL cation. Next, the bromide intermediate undergoes hydro-debromination over Pd/C in the presence of H$_2$, yielding HBr and alkyl-terminated species. The subsequent acidification of carboxylate anion with the generated HBr produces the carboxylic acid-terminated species, and [BMMIm]Br is regenerated simultaneously. Following similar procedures, the C$_{alkoxy}$-O bonds linked to each ester group are broken continually, resulting in the formation of the desired carboxylic acid. In this pathway, the IL achieves the breakage of the C$_{alkoxy}$-O bond in polyester, and Pd/C catalyzes the hydrogenation of the generated bromide intermediates. The IL and Pd/C co-catalyze the degradation of the polyester into carboxylic acid.

## Recycling polyesters with $\beta$-H over [P$_{4444}$]Br under the N$_2$ atmosphere

In the control experiment of MB treated with [BMMIm]Br under the N$_2$ atmosphere, methyl bromide was detected in the gas phase. Considering that the halide anion can function as a base to catalyze $\beta$-elimination reactions effectively, that is the dehydrohalogenation of halides, we propose a second strategy to directly deconstruct the polyesters with $\beta$-H using the ILs with halide anions as shown in Fig. 5. Tetrabutylphosphonium bromide ([P$_{4444}$]Br) was found to show the best performance (Supplementary Fig. 12), which was selected as the solvent and catalyst for deconstructing polyesters with $\beta$-H. Excitingly,

poly(β-hydroxybutyrate) (PHB) was favorably decomposed into crotonic acid in a yield of 65% with a small amount of 3-butenoic acid (6%), which are unsaturated organic acids. Interestingly, poly(butylene terephthalate) (PBT), poly(1,4-butylene succinate) (PBS), and poly(1,4-butylene adipate) (PBA) were deconstructed into terephthalate acid (TPA), succinic acid and adipic acid, respectively, accompanied with 1,3-butadiene. Furthermore, the co-polyester Poly(butylene adipate-co-terephthalate) (PBAT) was degraded into TPA, adipic acid, and 1,3-butadiene. Besides, the degradation of a polyester mixture consisting of PHB and PBT was carried out, and crotonic acid, 3-butenoic acid, TPA, and 1,3-butadiene were successfully produced as expected. This strategy achieves the degradation of polyesters with β-H into olefine acids and alkenes without additional hydrogen sources, enriching the variety of degradation products.

### Mechanistic studies on recycling polyesters with β-H over [P$_{4444}$]Br under the N$_2$ atmosphere

In order to explore the dehalogenation mechanism of halide over [P$_{4444}$]Br, control experiments were performed. It was indicated that benzyl bromide had no reactivity (Supplementary Fig. 13A), while bromoethane was transformed into ethylene under the same conditions (Supplementary Figs. 13B, 14). These results demonstrate that the IL could only catalyze dehydrohalogenation of halides with β-H, which is in accordance with the results reported[39]. However, no acetylene was detected in the gaseous phase when 1,2-dibromoethane was examined (Supplementary Fig. 13C), which reflects that the catalytic dehalogenation capability of the ILs with halide anions is limited.

Based on the above results and analysis, we propose a reasonable reaction pathway for the deconstruction of polyesters with β-H over [P$_{4444}$]Br under the N$_2$ atmosphere (Supplementary Fig. 13D). The C$_{alkoxy}$-O bond is firstly broken by the Br$^-$ anion, generating the IL cation-stabilized carboxylate anion and bromide intermediate. Under the catalysis of the Br$^-$ anion in quick succession, the bromide intermediate undergoes a dehydrobromination process, yielding HBr and alkenyl species. Subsequently, in situ acidification of carboxylate anion with the generated HBr proceeds, in which the carboxylic acid-

terminated intermediate is produced with the regeneration of [P$_{4444}$]Br. Following similar procedures, the IL breaks the C$_{alkoxy}$-O bonds in polyester and catalyzes dehydrobromination continuously, leading to corresponding olefine acids finally. In this protocol, the IL as the catalyst exhibits both nucleophilicity and basicity to accomplish the degradation of polyesters under metal-free conditions. In this sense, it can be foreseeable that it could contribute to reducing cost, which is crucial to polyester recycling in practice.

## Discussion

The findings described herein showcase a general strategy for recycling polyesters into carboxylic acids and hydrocarbons via breaking the C$_{alkoxy}$-O bond in polyesters using the ILs with halide anions. It is indicated that the hydrogen-bonding interaction between IL and ester group in polyester enhances the nucleophilicity of halide anion and activates the C$_{alkoxy}$-O bond. Among the effective ILs, [BMMIm]Br showed the best performance combined with hydrogenation over Pd/C, which could achieve the deconstruction of various polyesters including aromatic and aliphatic ones, copolyesters, and polyester mixtures into corresponding carboxylic acids and alkanes. In the absence of an additional hydrogen source, [P$_{4444}$]Br could achieve direct decomposition of the polyesters with β-H into carboxylic acids and alkenes under metal-free conditions. The strategy developed in this work paves the way to produce carboxylic acids and hydrocarbons from waste polyester plastics.

## Methods

### General procedures for the degradation of polyesters over [BMMIm]Br-Pd/C under the H$_2$ atmosphere

The decomposition reactions were performed in a stainless-steel autoclave equipped with a Teflon inner tube (16 mL inner volume) and a magneton. In general, polyesters, [BMMIm]Br, Pd/C (5 mg, 5 wt% Pd) were loaded in the autoclave in a glovebox under the N$_2$ atmosphere, and then the autoclave was sealed. H$_2$ was charged into the autoclave up to 5 MPa at room temperature. The autoclave was subsequently moved into a furnace at the desired temperature (e.g., 180 °C), and taken out to cool naturally after desired reaction time. The liquid

**Fig. 4 | Mechanism study on recycling polyesters over [BMMIm]Br-Pd/C under the H$_2$ atmosphere.** (**A, B**) Control experiments for the reaction of MB in the presence of H$_2$ or N$_2$. Reaction conditions: MB (1 mmol), [BMMIm]Br (2 mmol), Pd/C (5 mg, 5 wt% Pd), 5 MPa H$_2$ or 1 MPa N$_2$, 160 °C, 24 h; (**C**) The hydrogenolysis reaction of 3-bromobutyric acid. Reaction conditions: 3-bromobutyric acid (1 mmol), [BMMIm]Br (2 mmol), Pd/C (5 mg, 5 wt% Pd), 5 MPa H$_2$, 160 °C, 24 h; **D** Possible reaction pathway for the degradation of polyesters over [BMMIm]Br-Pd/C under the H$_2$ atmosphere. R=H or CH$_3$, $m \geq 0$.

**Fig. 5 | Deconstruction of various polyesters (R = aryl or alkyl groups, $m \geq 0$, $m_1 \geq 1$, $m_2 \geq 2$) with $\beta$-H over [P$_{4444}$]Br under the N$_2$ atmosphere.** [a]Reaction conditions: **a** polyester (1 mmol), [P$_{4444}$]Br (2 mmol), 1 MPa N$_2$, 200 °C, 8 h; **b** polyester (0.5 mmol), 12 h; **c** polyester (0.5 mmol), 10 h; **d** polyester (0.25 mmol), 12 h; **e** PHB (0.5 mmol), PBS (0.25 mmol), 10 h.

products and their yields were determined by $^{1}$H and $^{13}$C NMR spectroscopy, and the gaseous products were determined by GC or GC-MS.

### General procedures for the degradation of polyesters with $\beta$-H over [P$_{4444}$]Br under the N$_2$ atmosphere

The deconstruction reactions were performed in a stainless-steel autoclave equipped with a Teflon inner tube (16 mL inner volume) and a magneton. Typically, polyesters, [P$_{4444}$]Br were loaded in the autoclave in a glovebox under the N$_2$ atmosphere, and then the autoclave was sealed. N$_2$ was charged into the autoclave up to 1 MPa at room temperature. Subsequently, the autoclave was moved into a furnace at the desired temperature (e.g., 200 °C), and taken out to cool naturally after desired reaction time. The liquid products and their yields were determined by $^{1}$H and $^{13}$C NMR spectroscopy, and the gaseous products were determined by GC-MS.

### General procedures for the control experiments over [BMMIm]Br-Pd/C

The control experiments were performed in a stainless-steel autoclave equipped with a Teflon inner tube (16 mL inner volume) and a magneton.

For the reaction of MB over [BMMIm]Br-Pd/C under the H$_2$ atmosphere, MB (1 mmol), [BMMIm]Br (2 mmol), and Pd/C (5 mg, 5 wt% Pd) were loaded in the autoclave in a glovebox under the N$_2$ atmosphere, and then the autoclave was sealed. H$_2$ was charged into the autoclave up to 5 MPa at room temperature. Subsequently, the autoclave was moved into a furnace at 160 °C, and taken out to cool naturally after 24 h. The liquid products were determined by $^{1}$H and $^{13}$C NMR spectroscopy, and the gaseous products were determined by GC.

For the reaction of MB over [BMMIm]Br under the N$_2$ atmosphere, MB (1 mmol), [BMMIm]Br (2 mmol), and Pd/C (5 mg, 5 wt% Pd) were added in the autoclave in a glovebox under the N$_2$ atmosphere, and then the autoclave was sealed. N$_2$ was charged into the autoclave up to 1 MPa at room temperature. The autoclave was subsequently moved into a furnace at 160 °C, and taken out to cool naturally after 24 h. The liquid products were determined by high-resolution electrospray ionization mass spectrometry (HR-ESI-MS), and the gaseous products were determined by GC-MS.

For the reaction of 3-bromobutyric acid over [BMMIm]Br-Pd/C under the H$_2$ atmosphere, 3-bromobutyric acid (1 mmol), [BMMIm]Br (2 mmol) and Pd/C (5 mg, 5 wt% Pd) were loaded in the autoclave in a glovebox under the N$_2$ atmosphere, and then the autoclave was sealed. H$_2$ was charged into the autoclave up to 5 MPa at room temperature. Subsequently, the autoclave was moved into a furnace at 160 °C, and taken out to cool naturally after 24 h. The liquid products were determined by $^{1}$H and $^{13}$C NMR spectroscopy.

### General procedures for the control experiments over [P$_{4444}$]Br under the N$_2$ atmosphere

The control experiments were performed in a stainless-steel autoclave equipped with a Teflon inner tube (16 mL inner volume) and a magneton.

For the reaction of benzyl bromide over [P$_{4444}$]Br under the N$_2$ atmosphere, benzyl bromide (1 mmol) and [P$_{4444}$]Br (2 mmol) were added in the autoclave in a glovebox under the N$_2$ atmosphere, and then the autoclave was sealed. N$_2$ was charged into the autoclave up to 1 MPa at room temperature. The autoclave was subsequently moved into a furnace at 200 °C, and taken out to cool naturally after 12 h. The liquid products were determined by $^{1}$H and $^{13}$C NMR spectroscopy.

For the reaction of bromomethane over [P$_{4444}$]Br under the N$_2$ atmosphere, bromomethane (1 mmol) and [P$_{4444}$]Br (2 mmol) were added in the autoclave in a glovebox under the N$_2$ atmosphere, and then the autoclave was sealed. N$_2$ was charged into the autoclave up to 1 MPa at room temperature. The autoclave was subsequently moved into a furnace at 200 °C, and taken out to cool naturally after 12 h. The gaseous products were determined by GC.

For the reaction of 1,2-dibromoethane over [P$_{4444}$]Br under the N$_2$ atmosphere, 1,2-dibromoethane (0.5 mmol) and [P$_{4444}$]Br (2 mmol) were added in the autoclave in a glovebox under the N$_2$ atmosphere, and then the autoclave was sealed. N$_2$ was charged into the autoclave up to 1 MPa at room temperature. The autoclave was subsequently moved into a furnace at 200 °C, and taken out to cool naturally after 12 h. The gaseous products were determined by GC.

### NMR measurements
NMR spectra were recorded on Bruker Avance 400 or III 500 WB spectrometer equipped with 5 mm pulsed-field-gradient (PFG) probes. Chemical shifts were given in ppm relative to tetramethylsilane (TMS).

For the $^1$H and $^{13}$C NMR analysis performed at room temperature, the samples were dissolved in D$_2$O or DMSO-d6 with 1,3,5-trioxane (1 mmol) or 1,3,5-trimethoxybenzene (0.5 mmol) as the internal standard, and chemical shifts were recorded on Bruker Avance 400.

For the $^1$H, $^{13}$C, $^{17}$O, and $^{35}$Cl NMR analysis performed at 90 °C, wilmad coaxial insert NMR tubes were used to eliminate the effect of solvent, in which the sample was loaded in the outer tube while DMSO-d6 or NaCl-D$_2$O solution was in the inner tube, and chemical shifts were recorded on Bruker Avance III 500 WB. Specifically, for the $^1$H and $^{13}$C NMR analysis, the pure samples (MB or [BMMIm]Cl) and the MB-[BMMIm]Cl mixture (MB/[BMMIm]Cl=1/3, molar ratio) were added in the outer tube separately, and DMSO-d6 was in the inner tube. For the $^{17}$O NMR analysis, MB and the MB-[BMMIm]Cl mixture (MB/[BMMIm]Cl=1/1, molar ratio) were added in the outer tube separately, and DMSO-d6 was in the inner tube. For the $^{35}$Cl NMR analysis, [BMMIm]Cl and the MB-[BMMIm]Cl mixture (MB/[BMMIm]Cl=1/3, molar ratio) were added in the outer tube separately, and the NaCl-D$_2$O solution was added in the inner tube.

### GC analysis
Gas chromatography (GC, HP 4890D) is equipped with a TCD detector using argon as the carrier gas. Generally, the gaseous products from the decomposition of polyesters over [BMMIm]Br-Pd/C under the H$_2$ atmosphere were collected using a gas bag and analyzed by GC.

### GC-MS analysis
Gas chromatography-mass spectrometry (GC-MS, 7890B-5977A) is equipped with a packed column HP-5 MS using argon as the carrier gas. Generally, the gaseous products from the degradation of polyesters were collected using a gas bag and analyzed by GC-MS.

Gas chromatography-mass spectrometry (GC-MS, SHIMADZU-QP2010) is equipped with a packed column DB-5 MS using argon as the carrier gas. The gaseous products from the reaction of MB over [BMMIm]Br-Pd/C under the N$_2$ atmosphere were collected using a gas bag and analyzed by GC-MS.

### HR-ESI-MS analysis
The liquid products from the reaction of MB over [BMMIm]Br-Pd/C under the N$_2$ atmosphere were dissolved in methanol and analyzed by high-resolution electrospray ionization mass spectrometry (HR-ESI-MS, Bruker FT-ICR-MS (Solarix 9.4T)). The ionization method and mode of detection employed were indicated for the corresponding experiment, and all masses were reported in atomic units per elementary charge ($m/z$) with an intensity normalized to the most intense peak.

### DFT calculations
All calculations were performed with the Gaussian 09 package. Geometry optimizations were carried out at the M06-2X/def2-TZVP level at 298.15 K.

Note: Our DFT data presented had some limitations, which did not include greater numbers of cations and anions to demonstrate the effect of cluster size, but nonetheless supported the experimental results as they presented.

## Data availability
All data supporting the findings of this study are available within the paper and its Supplementary Information files. All data is available from the corresponding author upon request.

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

## Acknowledgements
This work was financially supported by the National Natural Science Foundation of China (22233006, 21890761, and 22121002), and the Chinese Academy of Sciences (027GJHZ2022053MI).

## Author contributions
Z.L. directed the project and designed the experiments. W.Z. performed the experiments. W.Z. and F.Z. carried out the DFT calculations. Y.Z., R.L., M.T., X.C., Y.W., and F.W. collaboratively analyzed the data. Z.L., B.H., and W.Z. contributed to the writing of the manuscript.

## Competing interests
The authors declare no competing interests.
