## [Peer Review File · Nature Communications]

A general strategy for recycling polyester wastes into carboxylic acids and hydrocarbonsReviewers' Comments:

Reviewer #1:

Remarks to the Author:

The paper reports an interesting strategy for the chemical degradation of polyesters. Whilst I find the paper contains good results I think that the messages are not very clear and the presentation needs some significant work to showcase the results further. For example PLA - propionic acid; PHB - butyric acid; PGA - acetic acid. Why is it "interesting" that PLGA makes acetic/propionic acid? On pg there are a of acronyms and it is difficult to follow the text. What is the market for the carboxylic acids that are being produced? What is the carbon balance for these processes?

There are processes out there that use ILs to degrade polymers - e.g. Epps et al ACS Macro Lett 2023, 1058/ Green Chem, 2022, 4130. They also use PHB to make an acid (alkene version), with H₂ this would then make the same thing as in this paper. Thus, I have some reservations around the novelty of the approach and I think this needs to addressed in any revised version.

Reviewer #2:

Remarks to the Author:

The authors reported very nice work of a general strategy for upcycling polyesters into carboxylic acids and hydrocarbons via breaking the Calkoxy-O bond in polyesters using the ILs with halide anions. All experiments have been conducted carefully and the results are very interesting. Therefore, I strongly recommend that the paper should be published on this journal after considering the following issues.

(1) In this paper, the effects of temperature, pressure, time and the amounts of IL on the PGA decomposition were investigated (fig. S5). Is the yield of acetic acid already in equilibrium under the studied conditions? Can it be further improved through condition optimization?

(2) Is there any other byproduct produced during the process of combining ionic liquids with hydrogenation over Pd/C to realize the decomposition of various polyesters into corresponding carboxylic acids and alkanes?

(3) Is there a difference in the activity of different polyesters during the decomposition of polyester mixtures into corresponding carboxylic acids and alkanes?

Reviewer #3:

Remarks to the Author:

The authors describe a novel degradation pathway afforded by ionic liquids, applied to several different polyesters, generating a variety of different commodity chemicals.

Major notes:

The figures have some confusing formats, making it difficult to follow along with the sometimes vague descriptions in the text. Figure 1: It's not clear what the distinction in conditions are between the top and bottom reaction sequences. It indicates the proton comes from H₂ in the top scheme. Where does the proton come from in the bottom scheme. Why does it necessarily form an unsaturated olefin? This is not clearly described in the text or clearly indicated in the figure. What are R1 and R2 in this figure? There is no way to get to poly(glycolic acid) using the generic structure in Fig. 1, or PLA or many other aliphatic polyesters.

It is slightly confusing how the transformation from PLA (or PGA) into butyric acid or propionic acid qualifies as up cycling. By what measure are you suggesting that the "value" of the acids or degradation products are higher than the plastic themselves. This issue renders the suggested topic of the paper somewhat misleading. I think it qualifies as a nice example of diversifying polyesters into some different commodity plastics, and the mechanism for this is certainly interesting. But it should

be written in a more clear way without the misleading suggestions of up cycling. Further along these lines, the degradation of PET to terephthalic acid is not up cycling, it's just chemical recycling. The hydrocarbons being generated also are extremely inexpensive materials.

The main (generic) structure in Figure 2 (top right) of the polyester is incorrect. The connections in the repeating unit is incorrectly depicted (anhydride/ester). Nearly all of the chemicals depicted in Figure 2 cannot be argued as products of up cycling. They are nearly all commodity chemicals that are extremely inexpensive.

This entire story should be carefully reconsidered and reframed before being considered for publication. It is difficult to wade through it with the mistakes highlighted above.

All of the NMR and mass spectra in the supporting information should have the signals identified for the different protons. It is not enough to just show the structure, especially because you're using an internal standard. It should not be up to the reader to decipher the spectra and figure out which signals belong to which protons/carbons.

Minor notes: All the references are missing the years, which makes it a bit inconvenient to look them up.

There are many grammatical errors scattered throughout.

Does the degradation of PLA generate CO₂, CH₄, and H₂O? Under certain conditions it will hydrolyze to lactic acid. Which conditions are you referring to? This is a slightly unfair characterization, since there are also many efforts to recycle PLA and other polyesters.

I think the reference to up cycling plastics deserves a few more highly relevant citations, including:

<https://doi.org/10.1021/acs.macromol.2c02054>

<https://www.nature.com/articles/s41586-021-04350-0>

<https://www.science.org/doi/10.1126/science.abg4503>

<https://pubs.acs.org/doi/10.1021/jacsau.3c00544>

<https://onlinelibrary.wiley.com/doi/full/10.1002/pol.20220137>

Response to reviewers' comments

Reviewer #1 (Remarks to the Author):

General comment: The paper reports an interesting strategy for the chemical degradation of polyesters. Whilst I find the paper contains good results I think that the messages are not very clear and the presentation needs some significant work to showcase the results further.

Comment 1: For example PLA - propionic acid; PHB - butyric acid; PGA - acetic acid. Why is it "interesting" that PLGA makes acetic/propionic acid?

Answer: We thank the reviewer for the comment. The word "interesting" was replaced by "expectedly" in the revised manuscript.

Comment 2: On pg there are a of acronyms and it is difficult to follow the text.

Answer: We thank the reviewer for the comment. In the revised manuscript, the full names of all the abbreviations were provided.

Comment 3: What is the market for the carboxylic acids that are being produced?

Answer: We thank the reviewer for the comment.

It was reported that the global carboxylic acid market share was USD 10.12 billion in 2021 and predicted to be around USD 13.04 billion by 2026, with a compound annual growth rate (CAGR) of 5.2%. In the revised manuscript, we added the statement on the market for the carboxylic acids.

references:

1. Cauwenbergh, R., Goyal, V., Maiti, R., Natte, K. & Das, S. Challenges and recent advancements in the transformation of CO₂ into carboxylic acids: straightforward assembly with homogeneous 3d metals. *Chem. Soc. Rev.* **51**, 9371-9423 (2022).

2. Global Carboxylic Acid Market (2021-2026) by Product Type, End-User, Geography, Competitive Analysis and the Impact of Covid-19 with Ansoff Analysis, <https://www.researchandmarkets.com/reports/5410359/global-carboxylic-acid-market-2021-2026-by> (accessed on 11 June 2022).

Acetic acid:

The global market size for acetic acid is expected to reach \$319.6 million by 2030. The global demand for virgin acetic acid was estimated to be 16.1 million tons in 2020, and it is projected to reach 19.6 million tons by 2027. Global market analysis forecast a compound annual growth rate (CAGR) around 3% for the period 2020–2027.

references:

1. Chen, Y. *et al.* Sustainable production of formic acid and acetic acid from biomass. *Mol. Catal.*

545 (2023).

2. Martin-Espejo, J. L., Gandara-Loe, J., Odriozola, J. A., Reina, T. R. & Pastor-Perez, L. Sustainable routes for acetic acid production: Traditional processes vs a low-carbon, biogas-based strategy. *Sci. Total Environ.* **840** (2022).

Propionic acid:

The annual global propionic acid production is estimated at over 450,000 tons with an annual growth rate of 2.7%. Its market price is currently \$1000 per ton for chemically produced and \$2000–3000 per ton for biologically produced propionic acid.

references:

1. Ammar, E. M. & Philippidis, G. P. Fermentative production of propionic acid: prospects and limitations of microorganisms and substrates. *Appl. Microbiol. Biot.* **105**, 6199–6213 (2021).

Butyric acid:

The worldwide market of butyric acid is approximately 80,000 tons per year at a price of ~\$1800 per ton.

references:

1. Jiang, L. *et al.* Butyric acid: Applications and recent advances in its bioproduction. *Biotechnol. Adv.* **36**, 2101–2117 (2018).

2. Luo, H. *et al.* Recent advances and strategies in process and strain engineering for the production of butyric acid by microbial fermentation. *Bioresour. Technol.* **253**, 343–354 (2018).

3. Wang, J., Lin, M., Xu, M. & Yang, S.-T. in *Anaerobes in Biotechnology* Vol. 156 *Advances in Biochemical Engineering-Biotechnology*. 323–361 (2016).

Terephthalate acid (TPA):

The market price of purified TPA was reported to be 850 \$ per ton. The worldwide production of purified TPA was estimated at 50 million tons per year, with an annual increase of 6%.

references:

1. Iglesias, J. *et al.* Advances in catalytic routes for the production of carboxylic acids from biomass: a step forward for sustainable polymers. *Chem. Soc. Rev.* **49**, 5704–5771 (2020).

2. Pellis, A., Acero, E. H., Gardossi, L., Ferrario, V. & Guebitz, G. M. Renewable building blocks for sustainable polyesters: new biotechnological routes for greener plastics. *Polym. Int.* **65**, 861–871 (2016).

3. Purified Terephthalic Acid Price Trend, <https://www.fibre2fashion.com/market-intelligence/textile-market-watch/purified-terephthalic-acid-pta-price-trends-industry-reports/1/?gcode=1>, accessed 7 February 2020.

Succinic acid:

The market size of succinic acid in the last decade was approximately 40–50 kton of succinic acid per year. The price of succinic acid fermentation (\$2000–3000 per ton) is in the range of the petrochemical succinic acid.

references:

1. Iglesias, J. *et al.* Advances in catalytic routes for the production of carboxylic acids from biomass: a step forward for sustainable polymers. *Chem. Soc. Rev.* **49**, 5704–5771 (2020).

2. Cukalovic, A. & Stevens, C. V. Feasibility of production methods for succinic acid derivatives: a marriage of renewable resources and chemical technology. *Biofuels, Bioprod. Biorefin.* **2**, 505–529 (2008).

3. Chemicals from biomass: A market assessment of bioproducts with near-term potential. Technical Report NREL/TP-5100-65509, 2014, vol. 20.

4. BioConSepT Market potential-for selected platform-chemicals report1, https://www.igb.fraunhofer.de/content/dam/igb/en/documents/publications/BioConSepT_Market-potential-for-selected-platform-chemicals_ppt1.pdf, 2011.

5. From the sugar platform to biofuels and biochemicals. Final report for the European Commission Directorate-General Energy (ENER/C2/423-2012/SI2.673791), 2015.

Adipic acid:

The market price of adipic acid is in the range of \$1500–1700 per metric ton, and the worldwide production of adipic acid is approximately 2.6 million tons with a growing demand of 3–5% per year. High-purity fibre-grade adipic acid is used to produce nylon 6,6, while lower grade adipic acid is employed in the synthesis of polyurethanes.

references:

1. Iglesias, J. *et al.* Advances in catalytic routes for the production of carboxylic acids from biomass: a step forward for sustainable polymers. *Chem. Soc. Rev.* **49**, 5704–5771 (2020).

2. Beerthuis, R., Rothenberg, G. & Shiju, N. R. Catalytic routes towards acrylic acid, adipic acid and ϵ -caprolactam starting from biorenewables. *Green Chem.* **17**, 1341–1361 (2015).

3. Pellis, A., Acero, E. H., Gardossi, L., Ferrario, V. & Guebitz, G. M. Renewable building blocks for sustainable polyesters: new biotechnological routes for greener plastics. *Polym. Int.* **65**, 861–871 (2016).

Crotonic acid:

The 2021 selling price in Europe was about 15.4 USD per kg, while the selling price of crotonic acid produced via thermal degradation of the bio-based PHB obtained from a pure culture was estimated to be 7.80–11.05 USD per kg in 2014.

references:

1. Elhami, V., Hempenius, M. A. & Schuur, B. Crotonic Acid Production by Pyrolysis and Vapor Fractionation of Mixed Microbial Culture-Based Poly(3-hydroxybutyrate-co-3-hydroxyvalerate). *Ind. Eng. Chem. Res.* **62**, 916–923 (2023).

2. Mamat, M. R. Z., Ariffin, H., Hassan, M. A. & Zahari, M. A. K. M. Bio-based production of crotonic acid by pyrolysis of poly(3-hydroxybutyrate) inclusions. *J. Cleaner Prod.* **83**, 463–472 (2014).

Comment 4: What is the carbon balance for these processes?

Answer: We thank the reviewer. Regarding the carbon balance for polyester decomposition, it was approximately 100% for all the polyester decomposition under corresponding conditions. As shown in Fig. 2, the yields of target carboxylic acids from the decomposition of various polyesters were 83% - 95% because of the presence of some intermediates depending on the chemical structures of

the polyesters under the same conditions. For example, as for the aliphatic polyesters (e.g., PGA), it is difficult to obtain a 100% yield or carbon balance owing to the loss in product collection caused by the volatility of aliphatic carboxylic acids. As for the aromatic polyesters (e.g., PET), we were not deliberately pursuing the complete conversion of the polyesters which would require a longer reaction time.

Comment 5: There are processes out there that use ILs to degrade polymers - e.g. Epps et al ACS Macro Lett 2023, 1058/ Green Chem, 2022, 4130. They also use PHB to make an acid (alkene version), with H₂ this would then make the same thing as in this paper. Thus, I have some reservations around the novelty of the approach and I think this needs to be addressed in any revised version.

Answer: We thank the reviewer for the comment. The mentioned papers (e.g. Epps et al ACS Macro Lett 2023, 1058/ Green Chem, 2022, 4130) that reported the PHB decomposition catalyzed by the basic ILs (e.g. [EMIM][AcO]) were cited in the revised manuscript (reference 23, 30). In this work, the halide ILs can directly break the C_{alkoxy}-O bond in polyesters via the nucleophilic substitution of halide anions on the C_{alkoxy} atom, which are applicable to all polyesters. The basic ILs reported in the reference (Green Chem, 2022, 4130) achieved the C_{alkoxy}-O bond cleavage via β -elimination of PHB, which is only applicable to the polyesters with β -H. In our work, we provide a general strategy for recycling various polyesters into carboxylic acids and hydrocarbons.

Reviewer #2 (Remarks to the Author):

General comment: The authors reported very nice work of a general strategy for upcycling polyesters into carboxylic acids and hydrocarbons via breaking the C_{alkoxy}-O bond in polyesters using the ILs with halide anions. All experiments have been conducted carefully and the results are very interesting. Therefore, I strongly recommend that the paper should be published on this journal after considering the following issues.

Comment 1: In this paper, the effects of temperature, pressure, time and the amounts of IL on the PGA decomposition were investigated (fig. S5). Is the yield of acetic acid already in equilibrium under the studied conditions? Can it be further improved through condition optimization?

Answer: We thank the reviewer for the comment. From the reaction pathway, there are no chemical equilibrium reactions in the PGA decomposition process under the studied conditions, and the theoretical yield of acetic acid can reach 100%. Actually, it is difficult to collect all generated acetic acid because of its high volatility, which was supported by the fact that a trace amount of acetic acid was detected in the collected gaseous phase. Condition optimization hardly further improved the yield of acetic acid under our experimental conditions.

Comment 2: Is there any other byproduct produced during the process of combining ionic liquids with hydrogenation over Pd/C to realize the decomposition of various polyesters into corresponding carboxylic acids and alkanes?

Answer: We thank the reviewer for the comment. No byproduct was detected in the transformation of all polyesters over Pd/C under the H₂ atmosphere. It should be pointed out that in this work the degradation of polyesters was carried out under the anhydrous condition, and no water or carbon dioxide was generated. In the whole process to degrade polyesters, no hydrolysis and decarboxylation reactions occurred, excluding the formation of byproducts.

Comment 3: Is there a difference in the activity of different polyesters during the decomposition of polyester mixtures into corresponding carboxylic acids and alkanes?

Answer: We thank the reviewer for the comment. For the decomposition of different polyester mixture, there still exists an activity difference in different polyesters, mainly depending on their chemical structures and properties. This was supported by the fact that one polyester was preferentially decomposed completely into corresponding carboxylic acid, and the other decomposed partially.

Reviewer #3 (Remarks to the Author):

General comment: The authors describe a novel degradation pathway afforded by ionic liquids, applied to several different polyesters, generating a variety of different commodity chemicals.

Major notes:

Comment 1: The figures have some confusing formats, making it difficult to follow along with the sometimes vague descriptions in the text. Figure 1: It's not clear what the distinction in conditions are between the top and bottom reaction sequences. It indicates the proton comes from H₂ in the top scheme. Where does the proton come from in the bottom scheme. Why does it necessarily form an unsaturated olefin? This is not clearly described in the text or clearly indicated in the figure. What are R1 and R2 in this figure? There is no way to get to poly(glycolic acid) using the generic structure in Fig. 1, or PLA or many other aliphatic polyesters.

Answer: We thank the reviewer for the valuable comments. Fig. 1 was redrawn, which clearly shows the decomposition routes for two kinds of polyesters in the presence of H₂ or N₂.

Comment 2: It is slightly confusing how the transformation from PLA (or PGA) into butyric acid or propionic acid qualifies as up cycling. By what measure are you suggesting that the "value" of the acids or degradation products are higher than the plastic themselves. This issue renders the suggested topic of the paper somewhat misleading. I think it qualifies as a nice example of

diversifying polyesters into some different commodity plastics, and the mechanism for this is certainly interesting. But it should be written in a more clear way without the misleading suggestions of up cycling. Further along these lines, the degradation of PET to terephthalic acid is not up cycling, it's just chemical recycling. The hydrocarbons being generated also are extremely inexpensive materials.

Answer: We thank the reviewer for the valuable comments and explanation on 'recycling' and 'upcycling'. Considering the difference between "upcycling" and "recycling" and the results of our work, we changed 'upcycling' into 'recycling' or 'transformation' in the revised manuscript.

Comment 3: The main (generic) structure in Figure 2 (top right) of the polyester is incorrect. The connections in the repeating unit is incorrectly depicted (anhydride/ester). Nearly all of the chemicals depicted in Figure 2 cannot be argued as products of up cycling. They are nearly all commodity chemicals that are extremely inexpensive. This entire story should be carefully reconsidered and reframed before being considered for publication. It is difficult to wade through it with the mistakes highlighted above.

Answer: We thank the reviewer for the rigorous comment. Based on the above comment, we revised Figure 2 (top right), and the meaning of R was identified as R₂=aryl or alkyl groups.

Comment 4: All of the NMR and mass spectra in the supporting information should have the signals identified for the different protons. It is not enough to just show the structure, especially because you're using an internal standard. It should not be up to the reader to decipher the spectra and figure out which signals belong to which protons/carbons.

Answer: We thank the reviewer for the comment. As suggested, we marked the signals for the protons and carbons of products in all the NMR spectra in the revised supporting information.

Minor notes:

Comment 5: All the references are missing the years, which makes it a bit inconvenient to look them up. There are many grammatical errors scattered throughout.

Answer: We thank the reviewer for the careful review. In the revised manuscript, the years for all the references were provided in the reference list.

Comment 6: Does the degradation of PLA generate CO₂, CH₄, and H₂O? Under certain conditions it will hydrolyze to lactic acid. Which conditions are you referring to? This is a slightly unfair characterization, since there are also many efforts to recycle PLA and other polyesters.

Answer: We thank the reviewer for the comment. We agree with the reviewer that there are many

efforts to recycle PLA and other polyesters, which are mainly based on hydrolysis and alcoholysis to break the C_{acyl} – O bond in polyesters. In this work, the used ILs directly break the C_{alkoxy}-O bond in polyesters via the nucleophilic substitution of halide anions on the C_{alkoxy} atom, generating products different from those derived from hydrolysis and alcoholysis. For PLA degradation in this work, an anhydrous and oxygen-free environment was required, and no CO₂, CH₄ and H₂O were detected. In the revised manuscript, we addressed this reaction condition.

Comment 7: I think the reference to up cycling plastics deserves a few more highly relevant citations, including:

<https://doi.org/10.1021/acs.macromol.2c02054>

<https://www.nature.com/articles/s41586-021-04350-0>

<https://www.science.org/doi/10.1126/science.abg4503>

<https://pubs.acs.org/doi/10.1021/jacsau.3c00544>

<https://onlinelibrary.wiley.com/doi/full/10.1002/pol.20220137>

Answer: We thank the reviewer for the valuable suggestion. The references mentioned by the reviewer were cited in the revised manuscript (reference 8, 10-13).

Reviewers' Comments:

Reviewer #1:

Remarks to the Author:

The authors have written a rebuttal which cover some of my points, the comments about market are well done (although I do not think appear in the main paper in as much details, which is fine). I do think that the comment I made about novelty does not seem to have been addressed. What the authors have done is interesting but it does need to be addressed that other ILs have done part of this before. This process has not really been indicated in the revised version. This could be view as tagging on a hydrogenation (using a a very common approach) to polyester degradation step - which is known.

Reviewer #2:

Remarks to the Author:

The authors have made a commendable effort to address most of the reviewers' concerns within the available space. It is recommended to include some important phenomenon descriptions in the manuscript for the convenience of readers to understand clearly. For example, it is difficult to collect all generated acetic acid because of its high volatility, which affects the yield of acetic acid. Thus, condition optimization hardly further improved the yield of acetic acid under experimental conditions. Apart from this issue, I have no problem in recommending it for publication.

Reviewer #3:

Remarks to the Author:

There are repeating unit structures that are incorrect in Figures 2, 4, and 5. You have an anhydride/ester linkage as the repeating unit in Figure 2 and 5. In Figure 4 you have a peroxide repeating unit. This should be changed before publication. The rest of the comments have been adequately addressed in my opinion.

Response to reviewers' comments

Reviewer #1 (Remarks to the Author):

Comment 1: The authors have written a rebuttal which cover some of my points, the comments about market are well done (although I do not think appear in the main paper in as much details, which is fine). I do think that the comment I made about novelty does not seem to have been addressed. What the authors have done is interesting but it does need to be addressed that other ILs have done part of this before. This process has not really been indicated in the revised version. This could be view as tagging on a hydrogenation (using a very common approach) to polyester degradation step - which is known.

Answer: We thank the reviewer for the comment. As suggested, we emphasize the work that the reviewer referred in the revised manuscript.

Reviewer #2 (Remarks to the Author):

Comment 1: The authors have made a commendable effort to address most of the reviewers' concerns within the available space. It is recommended to include some important phenomenon descriptions in the manuscript for the convenience of readers to understand clearly. For example, it is difficult to collect all generated acetic acid because of its high volatility, which affects the yield of acetic acid. Thus, condition optimization hardly further improved the yield of acetic acid under experimental conditions. Apart from this issue, I have no problem in recommending it for publication.

Answer: We thank the reviewer for the valuable comment. As suggested, we add some important phenomenon descriptions in the revised manuscript.

Reviewer #3 (Remarks to the Author):

Comment 1: There are repeating unit structures that are incorrect in Figures 2, 4, and 5. You have an anhydride/ester linkage as the repeating unit in Figure 2 and 5. In Figure 4 you have a peroxide repeating unit. This should be changed before publication. The rest of the comments have been adequately addressed in my opinion.

Answer: We thank the reviewer for the valuable comments. We have made corrections to Fig. 2, 4 and 5 as the reviewer commented.